# Strategies for the Diagnosis of Granulocytic Anaplasmosis in Two Naturally Infected Dogs

**DOI:** 10.3390/ani14010049

**Published:** 2023-12-22

**Authors:** Gabriela-Victoria Martinescu, Larisa Ivănescu, Raluca Ștefănescu, Lavinia Andronic, Simona Mătiuț, Raluca Mîndru, Gheorghe Solcan, Liviu Miron

**Affiliations:** 1Faculty of Veterinary Medicine, Iasi University of Life Sciences, 8 Mihail Sadoveanu Alley, 700490 Iasi, Romania; martinescugabi11@yahoo.co.uk (G.-V.M.); raluca.stef@yahoo.ro (R.Ș.); biancalavinia7@gmail.com (L.A.); raluk_1990@yahoo.com (R.M.); gsolcan@uaiasi.ro (G.S.); livmiron@yahoo.com (L.M.); 2Praxis Medical Laboratory, 33 Independentei Boulevard, 700102 Iasi, Romania; simona.matiut@laboratorpraxis.ro

**Keywords:** *Anaplasma phagocytophilum*, dogs, qRT-PCR, IDEXX, FLEX4, blood donation

## Abstract

**Simply Summary:**

Anaplasmosis is a widely spread emerging tick-borne disease, generally transmitted by the *Ixodes* tick species. Two members of the *Anaplasma* genus—*A. phagocytophilum* (infecting neutrophils) and *A. platys* (infecting platelets)—cause granulocytic anaplasmosis and thrombocytopenia in some animals and humans. Our case report describes granulocytic anaplasmosis in two dogs routinely tested for blood donation. The blood smear was negative for tick-borne pathogens, and the haematology findings indicated thrombocytopenia in both dogs. The dogs were mildly positive in the ELISA tests for the detection of antibodies for *Anaplasma* spp. The qRT-PCR result was negative for *A. platys* but positive for *A. phagocytophilum*. In conclusion, the molecular test is mandatory for confirmation of *Anaplasma* spp. infection, and we suggest it should be implemented in all blood donors.

**Abstract:**

This study describes granulocytic anaplasmosis in two dogs naturally infected with *Anaplasma phagocytophilum*. The 3-year-old dogs (male and female) came from the same household and were referred to the Faculty of Veterinary Medicine in Iasi for blood donation. They were subject to standard routine tests: haematology blood test, blood smear, and serological tests (VETSCAN^®^ FLEX4 and IDEXX SNAP 4Dx Plus). The female dog had no medical problems, while the male dog experienced joint pain. The blood smear was negative for tick-borne pathogens, and the haematology findings indicated thrombocytopenia in both dogs, with the male dog also displaying eosinophilia. The two dogs were mildly positive in the ELISA tests for the detection of *Anaplasma* spp. antibodies; therefore, the blood samples were tested using the qRT-PCR method for *Anaplasma platys* and *Anaplasma phagocytophilum*. The qRT-PCR result was negative for *A. platys*, but it was positive for *A. phagocytophilum*. The treatment consisted of the administration of doxycycline for 28 days. In conclusion, the high number of cases with non-specific clinical signs, the different sensitivity and specificity of the immunochromatographic serological tests, as well as the possibility of confusing the morula during the cytological examination, make the molecular test mandatory for precise diagnosis.

## 1. Introduction

The genus *Anaplasma* includes Gram-negative bacteria of high importance for both veterinary and human health. Anaplasmosis is a widely spread emerging tick-borne disease, generally transmitted by *Ixodes* tick species, although other ways of transmitting the infection have also been reported [1,2]. Tick-borne agents can spread vertically in mammals during pregnancy, parturition, or in the postpartum stage. Blood transfusions are a further means of transmission, so it is important to analyse canine blood products in highly endemic areas [3,4,5]. Moving hosts and infected ticks on migrating mammals or birds may spread several tick-borne pathogens. Hence, *A. phagocytophilum* is able to survive between peaks of tick activity [6].

The screening for vector-borne infections in dogs is very important not only for veterinarians but also for public health and vector control monitoring as well [7]. Professional risk groups such as veterinarians, hunters, and farm workers, as well as the residents of endemic areas, are more predisposed to tick bites; hence, an increase in the number of cases of human anaplasmosis has been observed in these categories [8,9]. These professional categories are more exposed to tick biting because they are in direct contact with pets or wild animals, as well as located in areas with a high prevalence of ticks.

Two members of this genus, *Anaplasma phagocytophilum* (infecting neutrophils) and *Anaplasma platys* (infecting platelets), cause granulocytic anaplasmosis (canine and human) and canine cyclic thrombocytopenia, respectively [2,10].

In a worldwide meta-analysis on the prevalence of *A. phagocytophilum* in various animal and bird species, the prevalence identified in Europe was 19.91%, with a higher prevalence of anaplasmosis reported in wild animals (17.64%), compared to domestic animals (10.68%) [11]. Also, a 0.5–6.3% prevalence of granulocytic anaplasmosis has been reported in dogs [12]. In Europe, the seroprevalence of human anaplasmosis has been moderately reported compared to other areas of the world [8]. The prevalence of *A. phagocytophilum* detected in ticks collected from humans in Romania was low (5.5%) [13]. Even if this prevalence is low, the risk in Romania is generally moderate due to several factors (e.g., animal migration, global warming, etc.).

In Romania, several studies have been carried out regarding bacterial infection with *Anaplasma phagocytophilum* in animals, most of them being serological studies [14,15], followed by molecular studies [12,16,17]. Consequently, several animal species were tested, such as dogs [18], cattle [19,20], wild birds [21], red foxes [22], small mammals [23], and wild boar [17]. Concerning the granulocytic anaplasmosis agents, wild birds have a lower reservoir competence; also, foxes are a good model of sentinel species that can be suggestive of the risk posed to public health. Additionally, small mammals serve as significant reservoir hosts for a number of pathogenic agents and can also act as reservoirs for *Anaplasma phagocytophilum*, indicating their potential role in the epidemiology of anaplasmosis [21,22,23].

Because of the long period of antibody formation and multiple cross-reactions, the optimal diagnosis methods are considered molecular tests [24]. Following an active infection or pathogen contact, antibodies may be determined between 7 months and several years [25]. A false positive result occurring as a consequence of a cross-reaction following previous exposure to another *Anaplasma* sp. or *Rickettsia* is the main disadvantage of rapid serological multiplex screening tests. Therefore, it is recommended to confirm the diagnosis using another method. PCR testing on peripheral blood may be a more sensitive diagnosis method in infected dogs than serology because of the acute nature of granulocytic anaplasmosis. Asymptomatic animals infected with *Ehrlichia*, *Rickettsia*, or *Anaplasma* species may serve as natural reservoirs [1,26,27].

The objective of the study was to present diagnosis strategies for canine granulocytic anaplasmosis, a disease with non-specific symptoms or with an asymptomatic evolution in most cases. Molecular testing is regarded as the gold standard for vector-borne disease diagnosis, mainly in individuals without clinical signs.

## 2. Results

The microscopic observations of the entire blood smears (between 3 and 5 smears for each sample) were negative for tick-borne pathogens, including *Babesia* spp., *Anaplasma* spp., and *Ehrlichia* spp. The CBC results indicated thrombocytopenia in both dogs: male—platelets: 54 × 10^9^/L; female—12 × 10^9^/L (reference values 165–500 × 10^9^/L), with the male dog also displaying eosinophilia (2.36 × 10^9^/L, reference values 0–0.8 × 10^9^/L).

Following the abdominal ultrasound, splenomegaly was detected in both patients. The spleen presented a homogeneous appearance with specific echogenicity, regular margins, and normal vascularity (Figure 1).

The two dogs were mildly positive in the VETSCAN^®^ FLEX4 and IDEXX SNAP 4Dx Plus tests for the detection of *Anaplasma platys*/*Anaplasma phagocytophilum* antibodies, which is why the blood samples were further analysed using the Multiplex qRT-PCR method for the detection of *A. platys* and *A. phagocytophilum*. A specific amplification curve was obtained in both samples only for *A. phagocytophilum* (Figure 2). The Ct value obtained for the female dog (without clinical signs) was 19.35, which suggests an acute stage of infection, which was in opposition to the male dog (with joint pain), where the Ct value was 38.41, which might suggest a mild or chronic stage of the infection.

The treatment consisted of the administration of oral doxycycline, in doses of 10 mg/kg per day for 28 days. Spot-on afoxolaner and milbemycin oxime were prescribed to prevent tick infestation.

After 30 days, the dogs were brought for a control visit. No clinical signs were further observed by the owner. Therefore, a general examination and a CBC were performed, but no abnormalities were detected.

## 3. Discussion

According to the authors’ knowledge, this is the first study in the northeast of Romania in which the qRT-PCR method was used for the diagnosis of Canine Granulocytic Anaplasmosis, besides the usual methods (blood smear and serological assay).

Compared to blood smears or serological analyses, the qRT-PCR assay is more sensitive. Since detectable antibodies have not been produced yet, serologic tests are not very useful in the early acute phase of infection [28].

Dogs are considered sentinel species due to the similarity between human and canine *A. phagocytophilum* strains and because of the close relationship between humans and dogs. Dogs can also infest homes with infected ticks, increasing the incidence of human disease [12]. Human granulocytic anaplasmosis is usually seasonal with most cases observed during the period of high activity of ticks. Immunodepression, blood transfusion, and outdoor activities are major risk factors for developing *A. phagocytophilum* infection [2]. Screening for human anaplasmosis should be taken into consideration following the occurrence of animal anaplasmosis in a certain region [29].

The diagnosis of anaplasmosis can be challenging, particularly during the acute stage. The identification of *A. phagocytophilum* is not uncommon, taking into consideration that the main vector is *Ixodes ricinus*, one of the most widely spread tick species in Europe [10,30] and also in Romania [13,31,32,33]. Cytological blood smear analysis is an economical and rapid diagnosis method [29], but the limited number of circulating cells that can be effectively analysed by microscopy, a lack of infected cells, and the occurrence of intracellular artefacts that may imitate morulae make blood smear analysis difficult [28]. According to the literature, morulae in neutrophils are identified only in 7–32% of natural infection cases [26], which explains the fact that the blood smear was negative. Moreover, the morulae of *Anaplasma phagocytophilum* are very similar to the morulae of *E. ewingii* [26,34], requiring an additional diagnosis method besides the blood smear.

The two most significant and frequent clinical signs of canine tick-borne diseases appear to include thrombocytopenia and polyarthritis, which can be revealed in other diseases [35]. Polyarthritis (articular inflammations, joint rigidity, or lameness) is frequently reported in infections with *A. phagocytophilum*, *E. canis*, *E. ewingii*, or *B. burgdorferi* [26,36]. In Europe, lameness was detected in 2 out of 18 canine patients with granulocytic anaplasmosis [37]. We consider that information in the medical history regarding the origin of the polyarthritis, in concert with previous infestation with ticks, helps the clinician determine whether the problem is strictly orthopaedic or whether they should include tick-borne disease on the differential diagnostic list.

The most frequent laboratory finding in *A. phagocytophilum* infection is thrombocytopenia [1,34,38,39]. Thrombocytopenia is caused by immune-mediated and inflammatory processes, which are consequences of platelets sequestered in the spleen and destroyed by the immune system [26]. In addition, eosinophilia was reported occasionally in dogs with granulocytic anaplasmosis [38,40], which is also one of the results of our study.

Splenomegaly is a common finding of granulocytic anaplasmosis detected by using diagnostic imaging (abdominal radiograph or ultrasound) [3,26,40,41,42]. In natural infections with *A. phagocytophilum*, splenomegaly was revealed in 12–100% of dogs [1], according to our observation. Ultrasonographic assessment of the spleen size is subjective. In dogs, the spleen margins appear round, and the organ extends caudally and to the right side of the abdomen [43,44]. Splenomegaly with normal echogenicity is found secondary to extramedullary haematopoiesis (including *Babesia* spp. and *Anaplasma* spp. infections and immune-mediated anaemia), infectious diseases, malignant infiltrations (lymphoma, mast cell tumours), splenic torsion, congestion, etc. [43,44].

Seroprevalence has been used over time to obtain an overview of canine vector-borne diseases [45]. Also, this is a rapid and economical method, used especially in private veterinary clinics. Firstly, the blood samples were tested with VETSCAN^®^ FLEX4 (93.3% sensibility; 96.4% specificity; as specified by the manufacturer), but according to Liu et al. in 2018 [7], the sensibility degrees of VETSCAN^®^ FLEX4 and SNAP 4DX Plus test were 12.7% versus 84.5%, which is why we decided to also test the blood samples with the second serological test. This was performed in accordance with a serological study published in 2018, where 11 out of 1253 blood samples tested for donation were positive for *A. platys*/*A. phagocytophilum*, using the SNAP 4DX Plus test [46]. Nevertheless, the results for both tests were mildly positive, so we tested the samples for confirmation using qRT-PCR.

Molecular options are very ample and specific; thus, we decided to perform a qRT-PCR, which was considered a more sensitive diagnosis method, because serological tests could give cross-reactivity with other rickettsia bacteria [26]. In our opinion, in the case of the female dog, considering the results in the qRT-PCR, the threshold value (Ct) indicates acute granulocytic anaplasmosis. Other authors observed high sensitivity of the molecular tests, and positive results appeared in approximately six to eight days before the identification of morulae in the cytological blood smear, which has also been confirmed by our negative result in the examination of the blood smear [1]. Additionally, morulae of *Anaplasma* spp. can develop for 4–8 days after inoculation in dogs infected experimentally [37]. New molecular strategies, such as qRT-PCR for the diagnosis of vector-borne pathogens, allow early detection and adequate treatment, thus reducing the consequences of infections with other tick-borne diseases. Furthermore, multiple gene-targeting real-time molecular diagnosis techniques have been developed for the detection of vector-borne pathogens in blood, tissue, and vectors [29].

Doxycycline treatment has been successfully implemented in several studies in dogs and humans with natural *A. phagocytophilum* infection in Europe and the USA [1,8,47,48,49].

Veterinarians are progressively identifying dogs with co-infections of vector-borne diseases. The immune system of patients exposed to ticks (without apparent symptoms) may be impacted by stress, pregnancy, or immunosuppressive medications, which may cause tick-borne disease symptoms to occur or even aggravate. We consider that molecular screening for vector-borne diseases should be performed both in suspected and healthy dogs, especially in blood donors.

## 4. Materials and Methods

### 4.1. Clinical Presentation

Two 3-year-old dogs, a male and a female, from the same household, were referred to the Faculty of Veterinary Medicine in Iasi for blood donation. The medical records of the two dogs indicated that both had received full vaccinations, and neither of them had been treated with ectoparasiticides in the previous six months.

The clinical examination revealed no medical problems regarding the female dog, while the male dog experienced joint pain of unknown aetiology detected in all limbs.

### 4.2. Diagnostics

In addition to the general clinical examination, the standard protocol for donating blood also includes blood testing for vector-borne pathogens (haematology blood test, blood smear, and serological tests). Venous blood samples were collected from the cephalic vein in tubes with EDTA and tubes without anticoagulants. For the detection of *Babesia* piroplasms, thin blood smears were stained using the Diff-Quick kit (Merck, Millipore Sigma, St. Louis, MO, USA) and examined using an optical microscope at ×1000.

A complete blood count (CBC) was performed using a VetScan HM5 analyser (Abaxis, Zoetis, Parsippany, NJ, USA). In addition, for the detection of *Anaplasma platys*/*Anaplasma phagocytophilum*, *Ehrlichia* spp., and *B. burgdorferi* antibodies and the identification of the *D. immitis* antigen, two tests were used—VETSCAN^®^ FLEX4 (Zoetis, Parsippany, MO, USA) and IDEXX SNAP 4Dx Plus (IDEXX Laboratories, Inc., Westbrook, ME, USA) [50,51,52]. In addition, an ultrasound (Acuson NX3 Elite, Siemens, Berlin, Germany) was performed to evaluate the abdominal organs.

The molecular screening started with DNA extraction using BioMagPure 12 Plus (Biosan, Riga, Latvia). Genomic DNA was extracted from 200 µL whole blood using the Blood DNA Extraction Kit 200, according to the manufacturer’s protocol. The specific primers and probe mix for *A. platys* and *A. phagocytophilum* targeted 16S rDNA and msp2 genes, respectively, using the TaqMan principle for detection [53,54]. Therefore, for the detection of *A. platys,* the specific primers were EP-963F and EP-1029R, as well as a EP16Sp probe [53], and for the detection of *A. phagocytophilum,* the reaction was performed using ApMSP2F and ApMSP2R primers, and a ApMSP2P probe [54] (Table 1).

The reaction mixture for *A. platys* consisted of 12.5 μL of TaqMan Fast Advanced Master Mix (Thermo Fisher Scientific, Waltham, MA, USA), 140 nM of EP16Sp probe, 400 nM EP-963F and EP-1029R primers, 5 μL of template DNA, and nuclease-free water to a final volume of 25 μL. For *A. phagocytophilum*, the mixture included 12.5 μL of TaqMan Fast Advanced Master Mix (Thermo Fisher Scientific, Waltham, MA, USA), 125 nM of ApMSP2p probe, 900 nM ApMSP2F and ApMSP2R primers, 5 μL of template DNA, and nuclease-free water to a final volume of 25 μL. Negative controls (sterile water) were run beside the samples. An internal control was used to indicate the existence of genomic DNA and the absence of PCR inhibitors.

The multiplex amplification was performed in a C1000™ Thermal Cycler (Bio-Rad, Hercules, CA, USA), using the CFX96™ Real-Time Detection System. The thermal conditions consisted of 95 °C for 10 min, followed by 40 cycles of 95 °C for 15 s and 62 °C for 1 min. The fluorescent signal collected from FAM and HEX channels was analysed using CFX Manager Software Version 3.1 (Bio-Rad, Hercules, CA, USA).

## 5. Conclusions

Serological tests illustrate exposure to infection but not a precise acute infection or disease, whereas molecular tests are very specific and suggestive of a current infection. Also, blood samples are highly sensitive for the PCR detection of canine vector-borne diseases.

The information that the dogs belonged to the same household emphasises the idea that the prevalence of anaplasmosis in Eastern Romania is elevated and poorly diagnosed.

Canine Granulocytic Anaplasmosis is underestimated in veterinary clinics in Romania as a consequence of non-specific clinical signs, correlated with the different sensibility and specificity of the immunochromatographic serological tests, as well as the possibility of confusing the morula of *A. phagocytophilum* during the cytological examination of the blood. Therefore, *Anaplasma* spp. should be taken into consideration in differential diagnosis; it should be recommended to evaluate the association of symptoms, pathology, and molecular tests for a precise diagnosis, especially in dog blood donors.

## Figures and Tables

**Figure 1 animals-14-00049-f001:**
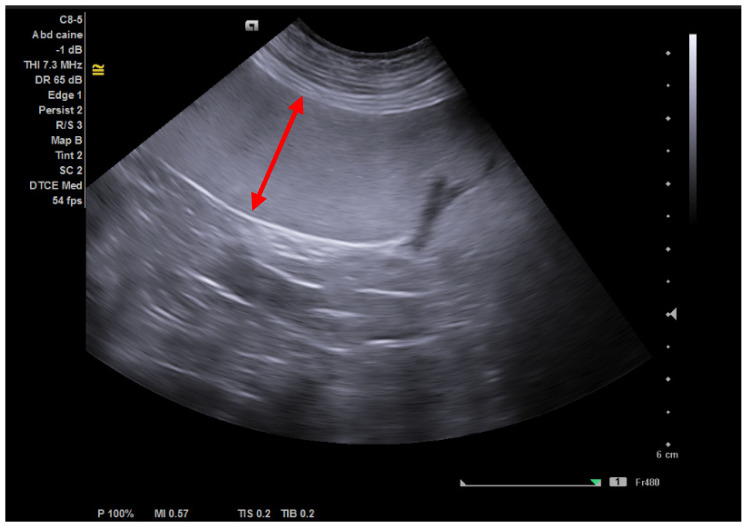
Splenomegaly; red arrow—homogeneous spleen.

**Figure 2 animals-14-00049-f002:**
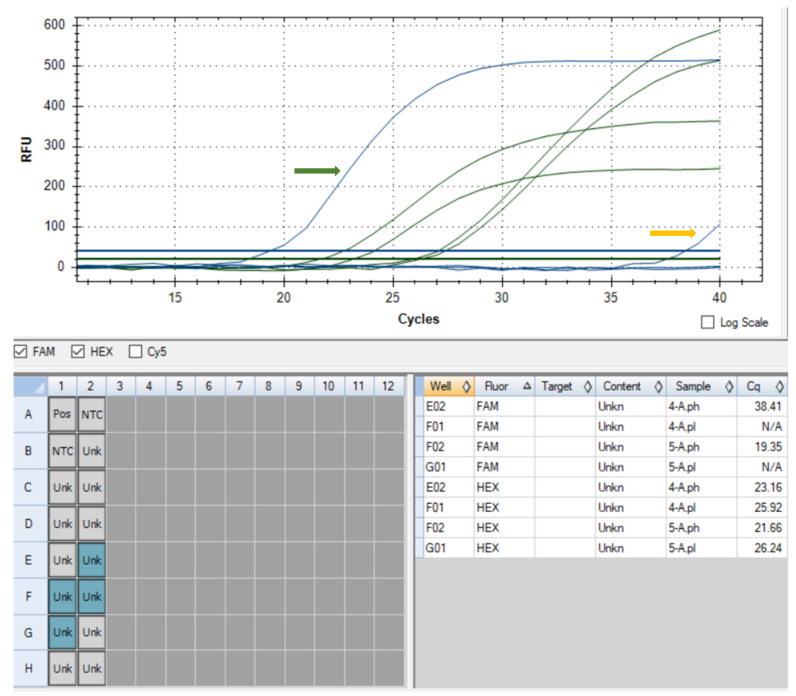
Amplification curves of blood samples for *A. platys* and *A. phagocytophilum* (green arrow—amplification curve for *A. phagocytophilum* in the female dog (Ct = 19.35); yellow arrow—amplification curve for *A. phagocytophilum* in the male dog (Ct = 38.41); RFU—relative fluorescence units).

**Table 1 animals-14-00049-t001:** Specific primers and probes used for the detection of *Anaplasma platys* and *Anaplasma phagocytophilum*.

Species	Primer/Probe	Sequence 5′–3′	Concentrations	References
*A. platys*	EP-963F	CGCAGTTCGGCTGGATCT	400 nM	[53]
	EP-1029R	CCCAACATCTCACGACACGA	400 nM	
	EP16Sp	FAM-TGACGACAGCCATGCAGCACCTG	140 nM	
*A. phagocytophilum*	ApMSP2F	ATGGAAGGTAGTGTTGGTTATGGTATT	900 nM	[54]
	ApMSP2R	TTGGTCTTGAAGCGCTCGTA	900 nM	
	ApMSP2p	FAM-TGGTGCCAGGGTTGAGCTTGAGATTG	125 nM	

## Data Availability

Data are contained within the article.

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
