# Peer review of "Strategies for the Diagnosis of Granulocytic Anaplasmosis in Two Naturally Infected Dogs"

_animals, 2023, doi:10.3390/ani14010049_

Round 1
Reviewer 1 Report
Comments and Suggestions for Authors
I found impossible to accurately review the manuscript because the references the authors provided are either misplaced, or non-existent. The paper, in its current form cannot be accepted unless the authors fix this major issue.
What do the authors mean by “Anaplasma species have the ability to resist between seasons in infected ticks from animals or migrating birds, which increases the expansion of pathogens”? Ticks are affected by climate, not the pathogens within them. This is not the main reason of the expansion of anaplasmosis. The reference (3) for this statement is not appropriate.
Change spread through neutrophils and spread through platelets with infecting neutrophils and infecting platelets.
“In Europe, the seroprevalence of human anaplasmosis has been moderately reported compared to other areas of the world [5, 10]”. Reference 10 does not match in this sentence.
“The prevalence of A. phagocytophilum detected in ticks collected from humans in Romania was low (5.5%) [11]”. I cannot find reference 11 anywhere.
The sentence “The identification of A. phagocytophilum is not uncommon, taking into consideration that the main vector is Ixodes ricinus, one of the most spread tick species in Europe” is repeated twice in the discussion, with different references, 7 and 23.
Comments on the Quality of English LanguageExcessive and sometimes unnecessary use of “therefore”.
Author Response
Thank you for your suggestions and you can find the new version of the article.

Reviewer 2 Report
Comments and Suggestions for Authors
Dear Authors your case report presents two cases of anaplasmosis in animals naturally infected with Anaplasma. The abstract adequately presents the case report subsequently described.
The introduction, briefly, describes the problem of Anaplasma both for its potential presence in domestic and wild animals, but also for the zoonotic risk for humans. The materials and methods shall describe reproducibly the techniques used and in particular the qRT-PCR which has been identified as the method of choice for the diagnosis of positive infection in the absence of clinical symptoms or in the presence of minimal signs.
As for the results, they are also rather short, while the discussion describes the different diagnostic possibilities for anaplasmosis, regarding the significance and sensitivity of the different methods.
I do not mind observing a part about the possible differential diagnosis performed at least in the male subject who had joint pain.
Please indicate if there has been any differential investigation.
The quality of the English language is proper
Regards
Author Response
Thank you very much for your suggestions and please find the new version of the article.

Reviewer 3 Report
Comments and Suggestions for Authors
I have thoroughly reviewed the article titled "Strategies for the Diagnosis of Granulocytic Anaplasmosis in Two Naturally Infected Dogs," focusing on the provided abstract. The study effectively describes granulocytic anaplasmosis in two naturally infected dogs, presenting relevant clinical information and diagnostic procedures. However, I recommend minor revisions for clarity and completeness.
Here are the minor points to optimize:
Keywords: Add two or three more keywords.
Introduction: 1. The authors should consider providing a brief explanation of alternative transmission routes for anaplasmosis, mentioned in line 32, to enhance reader understanding.
2. It would be beneficial if the authors further elaborate on why certain professional groups are more predisposed to tick bites (lines 38-40) to enhance the context and relevance of the statement.
3. The authors should elaborate on the significance or impact of the low prevalence of A. phagocytophilum in ticks collected from humans in Romania (lines 50-51), providing insights into potential public health implications.
4. A brief synthesis or overview of the key findings from previous studies in Romania regarding Anaplasma phagocytophilum infections in various animal species (lines 52-55) should be included by the authors.
5. The authors should offer further explanation on the challenges posed by the long period of antibody formation and multiple cross-reactions (lines 56-57).
6. The authors should conclude the introduction with a more explicit statement of the study's objective, emphasizing the importance of molecular tests for confirming vector-borne pathogen infections in dogs.
Results: 1. The blood smear results were negative for tick-borne pathogens, including Babesia spp., Anaplasma spp., and Ehrlichia spp. However, the authors should consider providing more context on the microscopic examination methodology to enhance clarity.
2. Thrombocytopenia was observed in both dogs, with the male dog's platelet count at 54 x 10^9 /l and the female's at 12 x 10^9 /l (reference values 165-500 x 10^9 /l). The authors must provide a more detailed discussion on the implications of thrombocytopenia in the context of granulocytic anaplasmosis.
3. While splenomegaly was detected in both dogs through abdominal ultrasound, the authors should provide additional information on the criteria used for diagnosing splenomegaly.
4. The authors must optimize Figure 2 by providing a detailed legend, axis labels, and any relevant annotations to ensure a comprehensive understanding of the amplification curves by the readers.
Discussion:
1. While the authors commendably employ the qRT-PCR method for diagnosing Canine Granulocytic Anaplasmosis, a more detailed discussion is needed on the advantages and limitations of this method compared to traditional approaches like blood smear and serological assays.
2. The authors mention that A. phagocytophilum identification is not uncommon, emphasizing the importance of dogs as sentinel species. However, they should elaborate on how this impacts public health, specifically in regions with close human-canine interactions and the potential for increased human disease incidence.
3. The discussion regarding the challenging diagnosis of anaplasmosis during the acute stage is insightful. However, the authors should further discuss the prevalence of morulae in neutrophils and the limitations of relying solely on blood smears for diagnosis, considering the similarity between morulae of A. phagocytophilum and E. ewingii.
4. The authors note the occurrence of polyarthritis in tick-borne diseases, particularly A. phagocytophilum infection. They should provide a more discussion on how clinicians can differentiate these diseases based on clinical signs and laboratory findings.
5. Thrombocytopenia, eosinophilia, and splenomegaly are discussed as common findings in A. phagocytophilum infection. However, the authors should elaborate on the clinical significance of these findings, especially in terms of differential diagnoses and treatment considerations.
6. While the authors touch upon the increasing recognition of co-infections, they should elaborate on the importance of routine screening for vector-borne diseases in both healthy and suspected dogs, providing recommendations for veterinarians in practice.
Comments on the Quality of English LanguageMinor editing of the English language is required.
Author Response

(The authors gave the same response as above.)

Reviewer 4 Report
Comments and Suggestions for Authors
The manuscript submitted for review deals with a case report aiming to present the significance of molecular tests for confirmation of vector-borne pathogens infections in dogs, mainly in dogs without clinical signs.
The approaches used to arrive at a confirmatory diagnosis based on the use of PCR-based assays to detect the genetic material of the pathogen present in a blood sample in a laboratory setting are nowadays the most convenient and recognized by convention. In addition, the drug used, Doxycyclin, to treat the animals that are chronically affected with canine granulocytic anaplasmosis is one the most effective and commonly commercially available. Thus, no novelty on the subject is found in the submitted manuscript.
For a review see, for example:
Carrade DD, Foley JE, Borjesson DL, Sykes JE. Canine granulocytic anaplasmosis: a review. J Vet Intern Med. 2009 23(6):1129-41. doi: 10.1111/j.1939-1676.2009.0384.x
Sainz, Á., Roura, X., Miró, G. et al. Guideline for veterinary practitioners on canine ehrlichiosis and anaplasmosis in Europe. Parasites Vectors 2015 8, 75. https://doi.org/10.1186/s13071-015-0649-0
Atif, F.A.; Mehnaz, S.; Qamar, M.F.; Roheen, T.; Sajid, M.S.; Ehtisham-ul-Haque, S.; Kashif, M.; Ben Said, M. Epidemiology, Diagnosis, and Control of Canine Infectious Cyclic Thrombocytopenia and Granulocytic Anaplasmosis: Emerging Diseases of Veterinary and Public Health Significance. Vet. Sci. 2021, 8, 312. https://doi.org/10.3390/vetsci8120312
The approach used to gather the data and arrive at a more precise diagnosis is worth mentioning, particularly when dogs are taken to a veterinary setting as blood donors. By combining the use of general clinical examination, followed by testing blood samples using several methods to detect the presence of potential tick-borne pathogens (hematology blood test, blood smear, serological tests, and qRT-PCR assay) authors claim this is the first study in the North-East of Romania in which a real-time PCR assay was used for the diagnosis of Canine Granulocytic Anaplasmosis.
Overall, the manuscript is well written, the laboratory approach concerning the analytical methods utilized is adequate, and sufficient technical details are provided to replicate the work.
A few items needing revisión were found by the reviewer:
There are no line numbers in the manuscript, making it difficult for the reviewer to signal precise corrections to be made to the text manuscript. Authors should use the journal´ Animals adequate template
In the results section, page 2, authors allude to figure 2, indicating that “the Ct value obtained for the female dog (without clinical signs) was 19.35 that suggest a massive bacterial infection”. Authors should then explain why being such a massive bacterial infection, no infected granulocytes were observed in the Giemsa-stained blood smears. Should it rather be defined as the acute phase of the disease with enough numbers of bacterial genomes to be picked up by the qRT-PCR?
In the legend to Figure 2, authors should indicate which curve/sample belongs to the male/female dog
In the discussion section, page 4, paragraph 4 authors use the statement “… in the sensibility between VETSCAN® FLEX4 and SNAP 4DX Plus test…” Authors should use “sensitivity” instead of “sensibility”.
Authors should carefully check the numbering of references. There are two number 1 references in the references section. If both references are to be used in the manuscript, which I think it is the case, citation numbering in the text should be renumbered consecutively from reference 2 onwards. Likewise, the references are to be renumbered in the references section
A reference number 46 is cited in the text. However, there is no reference number 46 in the references section
Comments on the Quality of English LanguageMinor editing of English language required
Author Response
Thank you very much for your suggestions, please find the new version of the article.

Round 2
Reviewer 1 Report
Comments and Suggestions for Authors
The revised manuscript has improved notably and can be accepted in its current format.